# MAGNET: Multi-granular Adaptive Gradient-guided Knowledge Distillation for Pareto-Efficient Tuning

## Abstract

Despite the remarkable advances achieved by large pretrained models (LPMs), their practical utility remains significantly constrained due to prohibitive computational and memory demands. While knowledge distillation (KD) partially alleviates this challenge, existing KD techniques rely predominantly on heuristically selected student architectures, resulting in suboptimal teacher-student pairings that frequently fail to achieve Pareto-optimal trade-offs between efficiency and performance. To overcome this limitation, we introduce MAGNET, a multi-granular adaptive gradient-guided knowledge distillation framework designed to analytically derive Pareto-efficient student architectures without heuristic intervention. Specifically, MAGNET initiates a concise gradient-profiling step on a small validation set, computing mean absolute gradients to rank layers according to their saliency. Based on this gradient-informed hierarchy, MAGNET selectively inherits only the most informative blocks, forming a highly compact student model. Within each selected block, MAGNET further masks parameters exhibiting minimal gradient magnitudes and executes a unified, single-stage training procedure integrating direct supervision, logit matching, and feature alignment. Comprehensive experiments across various vision and language benchmarks demonstrate that MAGNET consistently achieves superior accuracy with significantly fewer parameters and reduced computational overhead compared to state-of-the-art (SOTA) KD approaches.

## 1 Introduction

Large pretrained models represent a transformative shift in artificial intelligence, consistently achieving SOTA performance across diverse domains Dosovitskiy et al. (2020); Devlin et al. (2019). However, their substantial computational demands present a significant deployment barrier, restricting practical use primarily to data centers and limiting accessibility for resource-constrained platforms such as mobile devices and autonomous systems Cheng et al. (2017); Kim et al. (2024). This deployment gap not only poses practical limitations but also escalates sustainability concerns due to increasing energy requirements associated with large-scale inference Schwartz et al. (2020); Zhu et al. (2024). Knowledge distillation has emerged as a fundamental approach to model compression, aiming to transfer capabilities from large teacher models to smaller, efficient student models Hinton et al. (2015). The KD landscape has considerably evolved from initial logit-matching methods Hinton et al. (2015) to more sophisticated techniques leveraging intermediate features Romero et al. (2015), relational structures Park et al. (2019), and attention mechanisms Zagoruyko & Komodakis (2016).

Despite such advancements, KD is fundamentally constrained by the prevalent heuristic approach to student architecture selection, leading to inherently sub-optimal performance Gou et al. (2021). Mainstream techniques typically adopt one of two problematic strategies: heuristic layer and dimension scaling, which inaccurately assumes uniform functional roles across model layers Xu et al. (2019), or multi-stage pipelines employing aggressive structured pruning prior to distillation Chu et al. (2021), risking irreversible damage to critical transferable representations.

To improve upon heuristic selection, recent approaches utilize Neural Architecture Search (NAS), tailored specifically for DAS Wang & Du (2021). Despite progress, traditional NAS methods remain computationally intensive Elsken et al. (2019), and even gradient-based zero-cost NAS proxies that heuristically rank untrained architectures Abdelfattah et al. (2021) operate over external search spaces, whereas MAGNET uses gradients for intra-model analysis of a fixed teacher rather than searching over candidate students. These proxies commonly rely on metrics like parameter counts, FLOPs, or handcrafted similarity scores, which frequently correlate weakly with final student accuracy after distillation Ma et al. (2024). Thus, architectures deemed optimal by conventional proxies often fail to become effective students, underscoring a critical gap in explicitly quantifying the intrinsic distillability of student candidates—an essential yet overlooked factor in architecture selection.

To overcome these challenges, we propose MAGNET, a novel analytical framework designed to systematically derive optimal student architectures by deeply interpreting the teacher's learning dynamics. Inspired by the principle which parameter gradients precisely reflect immediate contributions to task-specific knowledge, MAGNET initiates a single gradient-profiling step, computing mean absolute gradients across the entire parameter set. This generates a comprehensive, data-driven blueprint informing architecture optimization at two synergistic levels: first, coarse-grained selection of layers demonstrating high cumulative gradients, effectively addressing inter-layer functional heterogeneity. And second, fine-grained pruning within these layers by masking parameters exhibiting minimal gradient signals. This approach conceptually parallels the identification of "Lottery Ticket" subnetworks crucial for maintaining performance Iurada et al. (2024). Ultimately, MAGNET unifies these insights into a Pareto-efficient optimization strategy, explicitly crafting compact yet potent student architectures optimized for enhanced knowledge distillation.

Our contributions can be summarized as follows:

- We propose MAGNET, a novel multi-granular knowledge distillation framework guided by gradient-based saliency profiling. By leveraging gradient profiling step over a small validation set, MAGNET facilitates a principled mechanism for selecting and compressing model components, reducing reliance on manual architectural design heuristics and avoiding computationally expensive neural architecture search.

- We introduce a coarse-grained inheritance strategy that ranks teacher model layers according to their saliency and transfers only the most informative ones to the student model. This approach ensures the preservation of structurally and semantically critical components, enabling the student to retain the teacher's core representational capacity while maintaining a compact architecture.

- We develop a fine-grained masking mechanism that suppresses low-saliency parameters within each inherited layer based on intra-layer gradient magnitude distributions. This layer-wise adaptive sparsification effectively reduces parameter redundancy, contributing to both model efficiency and generalization without impairing the representational integrity of the student model.

## 2 RELATED WORK

### 2.1 KNOWLEDGE DISTILLATION

KD compresses large models by transferring knowledge from a teacher to a compact student, aiming to reduce computational cost while preserving performance Hinton et al. (2015). Early methods minimize the Kullback-Leibler (KL) divergence between the output distributions of teacher and student, allowing the student to leverage the soft targets to generalize more effectively Hinton et al. (2015). Subsequent approaches extend KD by incorporating intermediate representations. FitNets align internal feature maps Romero et al. (2015), while AT Zagoruyko & Komodakis (2016) and FSP Park et al. (2019) focus on transferring attention and feature relationships. CRD Tian et al. (2020) employs contrastive losses to encode relational knowledge. Recent studies refine logit-based KD, DKD Zhao et al. (2022) splits the KD loss into target and non-target class components, identifying the latter as a key driver of performance. Logit standardization Sun et al. (2024b) mitigates teacher-student variance, improving distillation stability.

Despite these advances, KD methods typically assume fixed student architectures, leading to sub-optimal teacher-student alignment Gou et al. (2021). This "define-then-distill" paradigm constrains efficiency-performance trade-offs. To address this, recent work integrates NAS into DAS Bercovich et al. (2025), though at significant computational cost Elsken et al. (2019). To accelerate architecture search, Auto-DAS Sun et al. (2024a) introduces predictive proxies to estimate distillation performance without training.

## 2.2 Network Sparsification for Efficient Architecture

Network sparsification aims to eliminate non-essential components in neural networks, addressing the inefficiencies of heuristic-based student model design. Structured pruning methods, such as Network Slimming Liu et al. (2017), introduce sparsity by regularizing batch normalization scaling factors to remove less critical channels. However, the reliance on fixed pruning heuristics may lead to the loss of important features, adversely affecting knowledge transfer. The Lottery Ticket Hypothesis suggests that certain subnetworks within a larger model can maintain comparable performance when trained independently Frankle & Carbin (2018). Although this approach offers theoretical advantages, identifying such subnetworks typically requires extensive computation and relies on magnitude-based pruning. Recent methods like DSMoE Guo et al. (2024) propose dynamic sparse mixtures-of-experts that route inputs through adaptive blocks, enabling sparsification without permanent parameter removal. While this approach preserves model capacity and mitigates information loss, it primarily targets efficient inference rather than designing compact student models for distillation. Pruning at Initialization (PaI) methods improve efficiency by evaluating parameter importance before training. Techniques such as SNIP Lee et al. (2019), GraSP Wang et al. (2020), and SynFlow Tanaka et al. (2020) apply gradient-based saliency criteria to select weights. These methods reduce training cost compared to iterative pruning but often underperform on complex benchmarks. NAS offers an alternative for student design, with differentiable approaches like DARTS Liu et al. (2019) balancing performance and efficiency. However, NAS often depends on proxy metrics such as parameter count and FLOPs, which have limited correlation with student performance after distillation. Despite progress, current pruning and NAS strategies do not consistently yield effective student architectures. A data-driven method that explicitly aligns student design with teacher knowledge remains essential for advancing knowledge distillation.

## 3 Method

### 3.1 Preliminaries

Neural networks in resource-constrained settings require high accuracy with low computational cost. To address this, we develop a multi-granular architecture that combines coarse- and fine-grained feature representations for efficient inference. Knowledge distillation, which transfers knowledge from a large teacher model to a compact student model, underpins our training approach Hinton et al. (2015). The problem is to optimize a model's parameters to minimize distillation loss. Formally, given a dataset $\mathcal{D} = \{(x, y)\}$, where $x$ is an input and $y$ is its label, we solve

$$\min_{\theta_S} \mathbb{E}_{(x,y) \in \mathcal{D}} \left[ \mathcal{L}_{KD} \left( M_S(x; \theta_S), M_T(x; \theta_T), y \right) \right], \tag{1}$$

where $\mathcal{M}$ is the model parameterized by $\theta$, $\mathcal{L}_{KD}$ is the knowledge distillation loss function that incorporates the student model output $M_S(x; \theta_S)$, the teacher model output $M_T(x; \theta_T)$, and the ground-truth label $y$, and $\mathbb{E}$ denotes the expectation operator, computing the average loss over the dataset $\mathcal{D}$.

### 3.2 Multi-Granular Architecture Derivation

We propose a multi-granular method to develop an efficient student model architecture by systematically reducing the complexity of the teacher model while sustaining the performance. As presented in Figure 1, the MAGNET method employs three stages including Gradient-Guided Quantification, Coarse-Grained Inheritance and Fine-Grained Masking to construct the effective student model. The complete pipeline is detailed in Algorithm 1 in the Appendix.

### 3.2.1 GRADIENT-GUIDED SALIENCY QUANTIFICATION

To quantify the functional importance of model components during fine-tuning, we define the Layer Saliency $S(L_i)$ for each layer $L_i$, computed as the mean absolute gradient of all parameters within that layer, averaged over a validation set $\mathcal{D}_{\text{val}}$:

$$S(L_i) = \mathbb{E}_{(x,y)\sim\mathcal{D}_{\text{val}}} \left[ \frac{1}{|\Theta(L_i)|} \sum_{\theta_j \in \Theta(L_i)} \left| \frac{\partial \mathcal{L}}{\partial \theta_j} \right| \right], \tag{2}$$

where, $\Theta(L_i)$ denotes the set of parameters in layer $L_i$, and $\mathcal{D}_{\text{val}}$ is a small validation set used to estimate average gradient saliency. The saliency score $S(L_i)$, as defined in equation 2, quantifies the average contribution of parameters in layer $L_i$. The proposed framework, MAGNET, leverages parameter gradients to assess their contribution to task-specific knowledge. In the context of model training, the loss function $\mathcal{L}(\Theta)$ defines a high-dimensional Loss Landscape, where the gradient magnitude $\|\nabla_\theta \mathcal{L}\|$ for a parameter $\theta$ measures its local sensitivity to changes in the loss. Layers exhibiting large cumulative gradient magnitudes during training reside in steeper regions of the loss landscape, indicating significant functional adaptation to the target data distribution.

From a first-order perspective, let $\theta^{(i)} \in \mathbb{R}^{|\Theta(L_i)|}$ denote the parameter vector of layer $L_i$ and consider a small perturbation $\Delta^{(i)}$. A Taylor expansion of the distillation loss around $\theta^{(i)}$ gives

$$\mathcal{L}(\theta^{(i)} + \Delta^{(i)}) = \mathcal{L}(\theta^{(i)}) + \left\langle \nabla_{\theta^{(i)}}\mathcal{L}, \Delta^{(i)} \right\rangle + \mathcal{O}\big(\|\Delta^{(i)}\|_2^2\big). \tag{3}$$

If the perturbation is bounded in $\ell_\infty$, i.e., $\|\Delta^{(i)}\|_\infty \leq \varepsilon$, Hölder's inequality yields

$$\sup_{\|\Delta^{(i)}\|_\infty \leq \varepsilon} \left| \mathcal{L}(\theta^{(i)} + \Delta^{(i)}) - \mathcal{L}(\theta^{(i)}) \right| \lesssim \varepsilon \left\| \nabla_{\theta^{(i)}}\mathcal{L} \right\|_1, \tag{4}$$

so the layer-wise gradient $\ell_1$-norm, and hence the saliency score $S(L_i)$ in equation 2, can be interpreted as a task-specific proxy for the local Lipschitz constant of the distillation loss along layer $L_i$, providing a concise, first-order importance signal.

After computing saliency scores, layers are ranked based on their scores, allowing for informed selection of the most influential components for architectural refinement. This gradient-guided criterion reduces reliance on ad-hoc heuristic design choices by providing a principled, task-aware proxy for importance, yielding an efficient and high-performing student model tailored to the target task.

### 3.2.2 COARSE-GRAINED INHERITANCE

To construct a compact yet performant student model, we first identify and inherit the most informative structural components from the teacher model using a gradient-based block selection strategy. Given the computed saliency scores $\{S(L_i)\}_{i=1}^{N_T}$ for each of the $N_T$ layers in the teacher model, we define a permutation $\pi : \{1, \ldots, N_T\} \to \{1, \ldots, N_T\}$ such that:

$$S(L_{\pi(j)}) \geq S(L_{\pi(j+1)}), \quad \forall j \in [1, N_T - 1], \tag{5}$$

where $S(L_i)$ denotes the mean absolute gradient saliency of layer $L_i$. Based on this ordered ranking, we select the top-$N_S$ most salient layers to form the inheritance set:

$$\mathcal{L}_T^{\text{inherit}} = \{L_{\pi(j)} \mid j = 1, \ldots, N_S\}. \tag{6}$$

The student model is then initialized by directly copying the weights from the selected teacher layers. Specifically, the $j$-th student layer's parameters are initialized as:

$$\Theta_S^{(j)}[t = 0] \leftarrow \Theta_T^{(\pi(j))}, \quad \forall j \in \{1, \ldots, N_S\}, \tag{7}$$

where $\Theta_S^{(j)}$ and $\Theta_T^{(\pi(j))}$ represent the parameter sets of the student and teacher layers, respectively. The coarse-grained inheritance strategy focuses on transferring structurally significant layers with the highest gradient saliency. The average gradient magnitude $S(L_i)$ serves as a proxy for each layer's sensitivity to task-specific supervision, with higher values indicating stronger functional adaptation. By selecting only the most salient layers, this approach preserves components that contribute most to the teacher's predictive capacity. This gradient-informed criterion aligns with established attribution principles, where larger gradients imply greater influence on the output. Compared to heuristic or uniform scaling strategies, the coarse-grained selection yields a compact yet expressive architectural backbone, ensuring that the student model retains the most transferable representations while eliminating redundant structures.

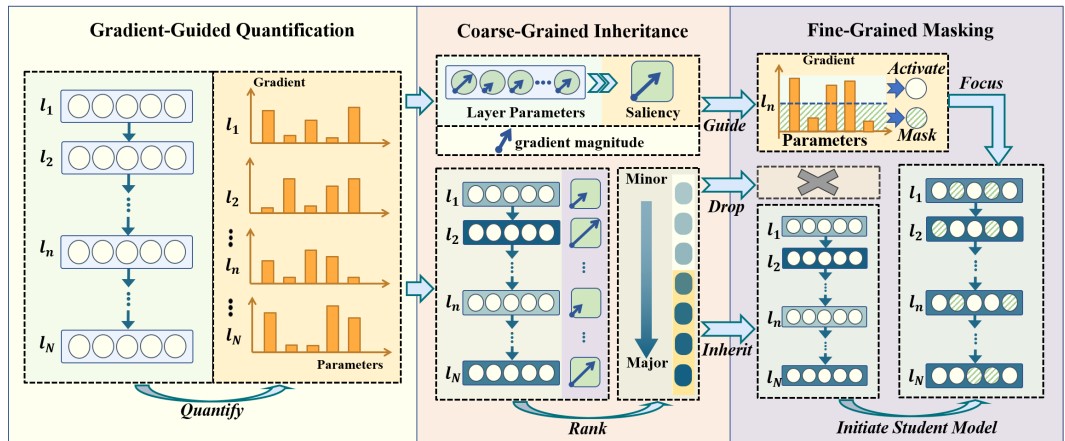

Figure 1: Overview of the MAGNET method's multi-granular architecture derivation process, comprising three stages: (1) **Gradient-Guided Quantification**: The first stage computes gradient-based saliency scores to evaluate the importance of each layer in the teacher model, identifying which components contribute most significantly to task performance. (2) **Coarse-Grained Inheritance**: Based on the saliency scores, high-impact layers are selected and directly transferred to the student model, preserving critical structural and functional knowledge with minimal adaptation. (3) **Fine-Grained Masking**: Within the inherited layers, less informative parameters are identified and masked, further reducing model complexity while maintaining predictive accuracy.

### 3.2.3 FINE-GRAINED MASKING

Following the coarse-grained inheritance of structurally salient layers, we introduce a fine-grained masking strategy to further improve the efficiency of the student model. Unlike conventional global pruning, our method performs parameter masking independently within each selected layer, allowing for localized adaptation to the unique functional roles of different layers. Let $\mathcal{L}_S = \{L_1, L_2, \ldots, L_{N_S}\}$ denote the set of inherited student layers. For each layer $L_\ell \in \mathcal{L}_S$, we denote its parameter set as $\Theta_S^{(\ell)} = \{\theta_k^{(\ell)}\}_{k=1}^{n_\ell}$, where $n_\ell = |\Theta_S^{(\ell)}|$ is the number of parameters in layer $L_\ell$. The saliency of each parameter $\theta_k^{(\ell)}$ is computed via

$$S_p(\theta_k^{(\ell)}) = \mathbb{E}_{(x,y)\sim\mathcal{D}_{\text{val}}}\left[\left|\frac{\partial\mathcal{L}}{\partial\theta_k^{(\ell)}}\right|\right], \tag{8}$$

yielding a set of saliency scores $\{S_p(\theta_k^{(\ell)})\}_{k=1}^{n_\ell}$ for each layer. These scores are sorted in ascending order within each layer:

$$S_p^{(\ell)}: \quad S_{(1)}^{(\ell)} \le S_{(2)}^{(\ell)} \le \cdots \le S_{(n_\ell)}^{(\ell)}. \tag{9}$$

To adapt sparsity to the functional importance of each layer while preserving a target global sparsity level, we derive an adaptive layer-wise masking ratio from the gradient-based layer saliency. Let $S^{(\ell)}$ denote the mean absolute gradient saliency of layer $L_\ell$. We first normalize these saliency values across the inherited layers:

$$\tilde{S}^{(\ell)} = \frac{S^{(\ell)} - \min_j S^{(j)}}{\max_j S^{(j)} - \min_j S^{(j)} + \varepsilon}, \quad \tilde{S}^{(\ell)} \in [0,1], \tag{10}$$

where $\varepsilon$ is a small constant to avoid division by zero. We then map the normalized saliency to a raw masking ratio in a prescribed range $[\gamma_{\min}, \gamma_{\max}] \subset (0,1)$:

$$\gamma_{\text{raw}}^{(\ell)} = \gamma_{\max} - \left(\gamma_{\max} - \gamma_{\min}\right)\tilde{S}^{(\ell)}, \tag{11}$$

so that highly salient layers ($\tilde{S}^{(\ell)} \approx 1$) receive a small masking ratio close to $\gamma_{\min}$, while less salient layers are pruned more aggressively toward $\gamma_{\max}$. To keep the overall sparsity aligned with a target

global masking ratio $\gamma_{\text{prop}} \in (0, 1)$, we rescale the raw ratios as

$$\bar{\gamma}_{\text{raw}} = \frac{1}{N_S} \sum_{\ell=1}^{N_S} \gamma_{\text{raw}}^{(\ell)}, \qquad \gamma_\ell = \text{clip}\left(\gamma_{\text{raw}}^{(\ell)} \cdot \frac{\gamma_{\text{prop}}}{\bar{\gamma}_{\text{raw}}}, 0, 1\right), \tag{12}$$

where $\text{clip}(\cdot, 0, 1)$ truncates the value into $[0, 1]$, and $\gamma_\ell$ is the final adaptive masking ratio for layer $L_\ell$.

Given the per-layer masking ratios $\gamma_\ell$, we compute the per-layer masking threshold $\tau_{\text{mask}}^{(\ell)}$ as

$$\tau_{\text{mask}}^{(\ell)} = S_{(q_\ell)}^{(\ell)} \tag{13}$$

where $q_\ell$ is the number of masked parameters in layer $L_\ell$ which is computed as:

$$q_\ell = \lfloor \gamma_\ell \cdot n_\ell \rfloor, \tag{14}$$

where $\lfloor \cdot \rfloor$ denotes the floor function. The masked parameter set for layer $L_\ell$ is then defined as

$$\Theta_{\text{mask}}^{(\ell)} = \left\{ \theta_k^{(\ell)} \in \Theta_S^{(\ell)} \mid S_p(\theta_k^{(\ell)}) < \tau_{\text{mask}}^{(\ell)} \right\}. \tag{15}$$

Finally, the complete set of masked parameters across the entire student model is given by the union over all layers:

$$\Theta_{\text{mask}} = \bigcup_{\ell=1}^{N_S} \Theta_{\text{mask}}^{(\ell)}. \tag{16}$$

The fine-grained masking stage therefore suppresses parameters with minimal gradient saliency under an adaptive, layer-aware sparsity schedule, which empirically exhibit weak task relevance and introduce representational redundancy. By assigning lower masking ratios to more salient layers and higher ratios to less active ones while matching a global sparsity budget, the method enforces localized sparsity tailored to intra-layer importance. This parameter-level refinement builds upon the coarse-grained structural selection by identifying the most functionally active subspaces within each inherited layer. Together, the two stages form a cohesive, gradient-guided compression framework that balances architectural compactness with task-specific expressiveness, thereby enhancing the overall efficacy of knowledge distillation.

## 3.3 UNIFIED OPTIMIZATION AND DISTILLATION STRATEGY

Once the student architecture is constructed, we train it using a unified single-stage procedure that integrates supervision, distillation, and structure-aware parameter updates. The total training objective is defined as:

$$\mathcal{L}_{\text{total}} = \alpha \cdot \mathcal{L}_{\text{CE}} + \beta \cdot \mathcal{L}_{\text{KL}} + \gamma_{\text{loss}} \cdot \mathcal{L}_{\text{Feat}}, \tag{17}$$

where $\mathcal{L}_{\text{CE}}$ is the cross-entropy loss between student predictions and ground-truth labels, and $\mathcal{L}_{\text{KL}}$ is the softened KL divergence between teacher and student logits, following the original Hinton KD formulation Hinton et al. (2015). The feature-level loss $\mathcal{L}_{\text{Feat}}$ encourages intermediate representation alignment. For each inherited student layer $L_j^S$, we compute the cosine similarity loss as:

$$\mathcal{L}_{\text{Feat}}^{(j)} = \begin{cases} 1 - \cos\left(f_j^S, f_j^T\right), & \text{if } j < N_S, \\ 1 - \cos\left(f_{N_S}^S, f_{\text{last}}^T\right), & \text{if } j = N_S, \end{cases} \tag{18}$$

where $f_j^S$ and $f_j^T$ denote the feature outputs of the student and teacher layers, respectively, and $f_{\text{last}}^T$ is the output of the teacher's last layer. The cosine similarity $\cos(a, b)$ between two feature vectors $a$ and $b$ is defined as:

$$\cos(a, b) = \frac{\langle a, b \rangle}{\|a\|_2 \cdot \|b\|_2}, \tag{19}$$

where $\langle \cdot, \cdot \rangle$ denotes the dot product and $\| \cdot \|_2$ denotes the Euclidean norm. The complete $\mathcal{L}_{\text{Feat}}$ is obtained by averaging over all layers:

$$\mathcal{L}_{\text{Feat}} = \frac{1}{N_S} \sum_{j=1}^{N_S} \mathcal{L}_{\text{Feat}}^{(j)}. \tag{20}$$

To implement structure-aware optimization, we apply a binary mask to control parameter updates. For each parameter $\theta_j$, its mask $m_j$ is:

$$m_j = \begin{cases} 0 & \text{if } \theta_j \in \Theta_{\text{mask}}, \\ 1 & \text{otherwise.} \end{cases} \tag{21}$$

This mask is applied to gradients during each optimization step to suppress updates to masked parameters.

## 4 EXPERIMENTS

### 4.1 EXPERIMENTAL SETUP

#### 4.1.1 DATASETS AND MODELS

We evaluate MAGNET across image classification (ImageNet-1k Deng et al. (2009), CIFAR-100 Krizhevsky et al. (2009), Food-101 Bossard et al. (2014), Stanford Cars Krause et al. (2013)), object detection (Tiny COCO Lin et al. (2014), PASCAL VOC 2012 Everingham et al. (2010)), and language understanding (IMDb Maas et al. (2011), SNLI Bowman et al. (2015), Hellaswag Zellers et al. (2019)). Robustness is assessed via ImageNet-C and CIFAR-100-C Hendrycks & Dietterich (2019). We adopt official data splits when available, defaulting to a 4:1 random partition otherwise. The teacher backbones include ViT-Base Dosovitskiy et al. (2020), ViTDet-B Li et al. (2022), and Bert-Base Devlin et al. (2019), with student architectures derived via our framework.

#### 4.1.2 BASELINES

We compare MAGNET against a suite of strong knowledge distillation methods. For vision tasks (covering both classification and detection), the baselines include FitNet Romero et al. (2015), AT Zagoruyko & Komodakis (2016), MGD Yang et al. (2022), NKD Yim et al. (2017), VITKD Yang et al. (2024), and WTTM Zheng & Yang (2024). For language tasks, our comparisons include Fit-Net, AT, MGD, DistilBERT Sanh et al. (2019), and WTTM.

#### 4.1.3 IMPLEMENTATION DETAILS

Experiments are conducted on NVIDIA A100 and H100 GPUs using the AdamW optimizer (batch size 64, weight decay 0.01). Unless otherwise specified, we distill 12-layer teacher backbones into 6-layer student architectures. All student models are trained for 50 epochs with an initial distillation learning rate of $3 \times 10^{-4}$. Regarding MAGNET, we set the adaptive masking bounds to $\gamma_{\min} = 0$ and $\gamma_{\max} = 1$ to fully leverage gradient saliency. For the gradient profiling step, we utilize a subset of 256 randomly selected samples from the training set; as detailed in Appendix B, this step incurs negligible computational overhead. For all comparative evaluations, we adopt a global masking ratio of $\gamma_{\text{prop}} = 0.4$, which we found empirically optimal.

### 4.2 MAIN RESULTS

We evaluate MAGNET on a broad suite of vision, robustness, detection, and language benchmarks, with results summarized in Table 1, Table 2, Table 3, and Table 4. On standard image classification, MAGNET yields the strongest student overall. Averaged over ImageNet-1k, CIFAR-100, Food-101, and Stanford Cars, the student attains 76.38% Top-1 accuracy and 92.81% Top-5 accuracy, surpassing the best competing distillation baseline NKD, which reaches 74.67% and 91.80%. The gains are most pronounced in data efficient regimes, where MAGNET achieves 83.94% and 77.16% Top-1 accuracy on CIFAR-100 and Food-101, and 73.82% on Stanford Cars, while remaining competitive on ImageNet-1k with 70.59% Top-1 compared with 72.31% for VITKD. To assess robustness, we further evaluate on ImageNet-C and CIFAR-100-C. MAGNET delivers the most robust student, with an average of 54.57% Top-1 and 73.27% Top-5 accuracy across the two corrupted benchmarks, improving over NKD, which obtains 53.12% and 73.20%. The advantage is particularly clear on CIFAR-100-C, where MAGNET reaches 69.72% Top-1 and 79.18% Top-5 accuracy compared with 66.19% and 78.51% for NKD, while remaining close to VITKD on ImageNet-C,

Table 1: PERFORMANCE COMPARISON WITH SOTA KNOWLEDGE DISTILLATION METHODS ON VISION DATASETS.

| Method | ImageNet-1k | | CIFAR-100 | | Food-101 | | Stanford Car | | Average | |
|---|---|---|---|---|---|---|---|---|---|---|
| | Top-1 | Top-5 | Top-1 | Top-5 | Top-1 | Top-5 | Top-1 | Top-5 | Top-1 | Top-5 |
| Teacher | 83.97 | 90.82 | 90.55 | 98.61 | 87.62 | 96.83 | 78.93 | 93.17 | 85.27 | 94.86 |
| DKD | 30.98 | 63.41 | 38.84 | 70.81 | 33.36 | 74.17 | 34.39 | 71.24 | 34.39 | 69.91 |
| FitNet | 31.86 | 64.19 | 39.39 | 70.38 | 34.91 | 74.28 | 33.57 | 71.51 | 34.93 | 70.09 |
| AT | 42.05 | 70.91 | 49.26 | 74.11 | 41.36 | 74.75 | 48.73 | 78.20 | 45.35 | 74.49 |
| MGD | 70.97 | 89.31 | 80.26 | 95.05 | 73.29 | 91.73 | 72.73 | 88.74 | 74.31 | 91.21 |
| NKD | 71.84 | **89.92** | 80.77 | 96.15 | 73.83 | 92.06 | 72.23 | 89.06 | 74.67 | 91.80 |
| VITKD | **72.31** | 89.68 | 78.14 | 93.53 | 74.06 | 92.17 | 73.54 | **90.37** | 74.51 | 91.44 |
| WTTM | 71.56 | 89.77 | 79.15 | 92.91 | 72.29 | 91.82 | 71.26 | 88.36 | 73.57 | 90.72 |
| **Ours** | 70.59 | 88.86 | **83.94** | **97.21** | **77.16** | **94.84** | **73.82** | 90.31 | **76.38** | **92.81** |

Table 2: PERFORMANCE ON CORRUPTED BENCHMARKS (IMAGENET-C and CIFAR-100-C).

| Method | ImageNet-C | | CIFAR-100-C | | Average | |
|---|---|---|---|---|---|---|
| | Top-1 | Top-5 | Top-1 | Top-5 | Top-1 | Top-5 |
| Teacher | 47.10 | 69.20 | 74.05 | 80.33 | 60.58 | 74.77 |
| DKD | 17.25 | 47.95 | 31.87 | 58.42 | 24.56 | 53.19 |
| FitNet | 18.03 | 48.67 | 32.41 | 57.96 | 25.22 | 53.32 |
| AT | 23.89 | 54.21 | 41.32 | 61.48 | 32.61 | 57.85 |
| MGD | 39.72 | 67.11 | 65.83 | 77.64 | 52.78 | 72.38 |
| NKD | 40.05 | 67.88 | 66.19 | 78.51 | 53.12 | 73.20 |
| VITKD | **40.68** | **68.12** | 63.47 | 76.92 | 52.08 | 72.52 |
| WTTM | 39.96 | 67.43 | 64.98 | 76.11 | 52.47 | 71.77 |
| **Ours** | 39.41 | 67.35 | **69.72** | **79.18** | **54.57** | **73.27** |

Table 3: PERFORMANCE COMPARISON ON OBJECT DETECTION TASKS.

| Method | Tiny COCO | | VOC 2012 | | Average | |
|---|---|---|---|---|---|---|
| | mAP | AP50 | mAP | AP50 | mAP | AP50 |
| Teacher | 44.11 | 70.04 | 48.90 | 74.69 | 46.50 | 72.37 |
| FitNet | 32.97 | 55.42 | 36.91 | 61.24 | 34.94 | 58.33 |
| AT | 34.52 | 57.03 | 38.47 | 62.73 | 36.50 | 59.88 |
| MGD | 38.73 | 64.38 | **42.37** | **69.31** | 40.55 | 66.85 |
| VITKD | 37.88 | 63.12 | 41.06 | 67.58 | 39.47 | 65.35 |
| WTTM | 38.95 | **66.21** | 41.52 | 67.02 | 40.24 | 66.62 |
| **Ours** | **40.21** | 65.67 | 41.69 | 68.26 | **40.95** | **66.97** |

Table 4: PERFORMANCE COMPARISON ON NATURAL LANGUAGE UNDERSTANDING TASKS.

| Method | IMDb | SNLI | Hellaswag | Average |
|---|---|---|---|---|
| Teacher | 92.87 | 89.26 | 40.40 | 74.18 |
| FitNet | 59.39 | 52.65 | 20.87 | 44.30 |
| AT | 66.13 | 55.46 | 22.71 | 48.10 |
| MGD | 88.29 | 79.56 | 31.97 | 66.61 |
| DistilBERT | 91.32 | **83.06** | 33.82 | 69.40 |
| WTTM | 87.18 | 81.43 | 32.24 | 66.95 |
| **Ours** | **93.02** | 82.67 | **35.87** | **70.52** |

where VITKD attains 40.68% Top-1 accuracy and MAGNET attains 39.41%. Beyond classification, MAGNET also transfers effectively to object detection. On Tiny COCO it achieves 40.21% mAP, and on VOC 2012 it reaches 41.69% mAP, which leads to the best student averages across the two detection benchmarks with 40.95% mAP and 66.97% AP50. On natural language understanding tasks, MAGNET attains an average accuracy of 70.52% across IMDb, SNLI, and HellaSwag, outperforming DistilBERT, which reaches 69.40%, and all other distillation baselines. On IMDb it obtains 93.02% accuracy, slightly exceeding the teacher at 92.87%, remains highly competitive on SNLI with 82.67% compared with 83.06% for DistilBERT, and achieves the highest student score on HellaSwag with 35.87%. Taken together, these results indicate that gradient guided architecture derivation and parameter masking yield compact student models that are accurate, robust, and broadly transferable across modalities and tasks.

### 4.3 ANALYSIS OF GRADIENT DYNAMICS

We conduct a detailed analysis of the teacher model's gradient distribution to validate the core design of MAGNET. This analysis examines both layer-level and parameter-level gradient behaviors to justify the use of gradient magnitude as a proxy for component importance. Figure 2(a) reports the average and standard deviation of absolute gradient magnitudes for the Query, Key, and Value components across all attention heads and layers in ViT-Base. The results reveal clear inter-layer and inter-head heterogeneity. Specifically, the Value components in mid-to-late layers exhibit the highest gradient activity, suggesting greater adaptation and task-specific relevance. This supports the coarse-grained inheritance mechanism, which prioritizes layers based on cumulative gradient saliency. Figure 2(b) visualizes the proportion of masked parameters per attention head, aggregated separately for Query, Key, and Value components. The masking decisions are based on intra-layer gradient ranking with a layer-wise threshold determined by a fixed global masking ratio. A consistent pattern emerges: Value components retain a lower ratio of masked parameters, while Query and Key components display higher redundancy. These findings validate the fine-grained masking strategy's ability to selectively suppress less informative weights while preserving critical substructures. Overall, the observed gradient dynamics reinforce the core assumptions of MAGNET. Both the magnitude and spatial distribution of gradients provide reliable signals for layer and parameter selection, enabling effective and structured student compression without heuristic tuning.

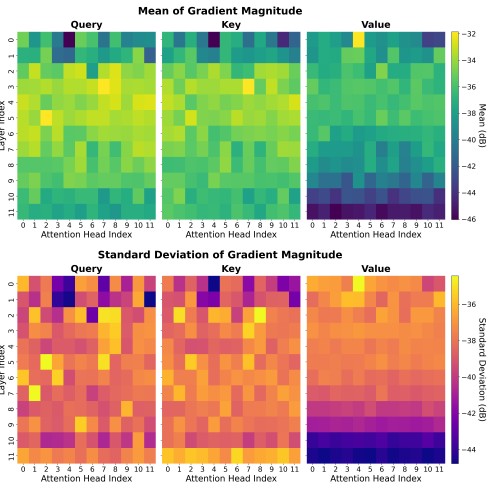
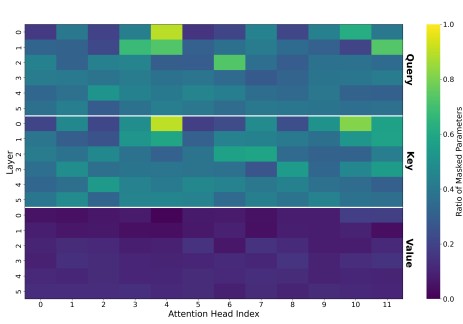

(a) Gradient magnitude statistics (Mean & Std).  (b) Masked parameter proportion per head.

Figure 2: Gradient dynamics analysis. (a) Statistics of gradient magnitudes reveal high activity in Value components. (b) The resulting mask density shows that informative components (Value) are preserved while redundant ones (Query/Key) are suppressed.

### 4.4 ABLATION STUDIES

We conduct extensive ablation studies to isolate and verify the contribution of each key component in MAGNET.

#### 4.4.1 IMPACT OF COARSE-GRAINED INHERITANCE

To demonstrate the superiority of our gradient guided layer selection, we compare MAGNET against three alternative strategies. Uniform selection chooses layers at evenly spaced depth positions as in DistilBERT Sanh et al. (2019). Random Init trains a student with the same depth pattern as MAGNET but without inheriting teacher parameters. Inverse-Grad deliberately inherits the layers with the lowest gradient saliency, using the same number of layers as MAGNET while reversing the saliency ranking. As reported in Table 5, our method consistently attains the highest Top-1 and Top-5 accuracy across CIFAR-100, Food-101, and Stanford Car. Uniform selection improves over Random Init, indicating that reusing teacher layers is beneficial even without saliency information, whereas Inverse-Grad performs worst among all strategies when it inherits the least salient layers on purpose. The clear ordering Ours greater than Uniform greater than Random Init greater than Inverse-Grad supports the view that gradient based saliency is not a superficial correlate of depth, but a meaningful signal of layer informativeness that must be used in the correct direction for effective knowledge distillation.

Table 5: ABLATION STUDY ON LAYER INHERITANCE STRATEGIES. OUR GRADIENT-GUIDED PROFILING IS COMPARED AGAINST UNIFORM, RANDOM-INIT, AND INVERSE-GRAD HEURISTICS.

| Selection Strategy | CIFAR-100 | | Food-101 | | Stanford Car | |
|---|---|---|---|---|---|---|
| | Top-1 | Top-5 | Top-1 | Top-5 | Top-1 | Top-5 |
| Uniform | 58.75 | 82.85 | 51.27 | 80.54 | 65.52 | 83.46 |
| Random Init | 49.80 | 78.43 | 42.31 | 70.35 | 52.96 | 81.48 |
| Inverse-Grad | 46.70 | 73.27 | 40.64 | 65.96 | 50.13 | 79.63 |
| **Ours** | **83.94** | **97.21** | **77.16** | **94.84** | **73.82** | **90.31** |

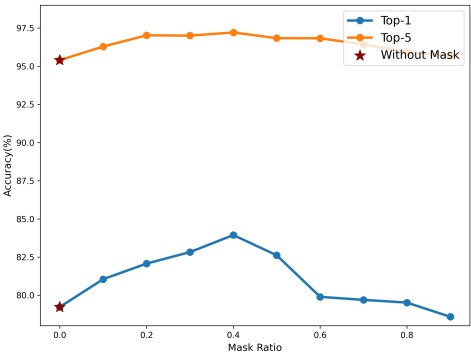 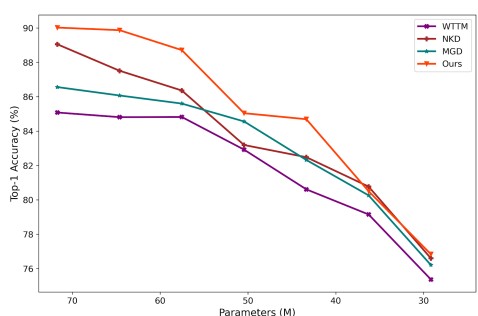

(a) Performance on CIFAR-100 under different fine-grained masking ratios. The red stars denote the baseline without masking ($\gamma_{\text{prop}} = 0$).

(b) Top-1 accuracy of student models distilled using different methods under varying model capacities.

Figure 3: Ablation study results. (a) shows the impact of varying the fine-grained masking ratio. (b) shows the scalability of MAGNET with different model capacities.

### 4.4.2 IMPACT OF FINE-GRAINED MASKING

We evaluate the effect of Fine-Grained Masking by varying the masking ratio $\gamma_{\text{prop}}$ from 0 (without masking) to 0.9 on the CIFAR-100 dataset. The results are shown in Figure 3(a). When $\gamma_{\text{prop}} = 0$, the student model preserves all inherited parameters. As the masking ratio increases, performance improves up to $\gamma_{\text{prop}} = 0.4$, where the Top-1 accuracy reaches 83.94% and Top-5 accuracy reaches 97.21%. Beyond this point, both metrics decline, indicating the detrimental impact of excessive pruning. These results demonstrate the contribution of Fine-Grained Masking in removing redundant parameters while preserving expressiveness. Moderate masking acts as a structural regularizer that enhances generalization and model compactness. The decline beyond $\gamma_{\text{prop}} = 0.4$ reflects the point at which critical parameters begin to be suppressed. This behavior supports the gradient-based saliency analysis shown in Figure 2(b), where importance-aware pruning enables a more optimal trade-off between model size and accuracy.

### 4.5 EFFECT OF MODEL CAPACITY

To evaluate the scalability of MAGNET under varying resource budgets, we compare its performance against WTTM, NKD, and MGD across a range of student model capacities. By varying the number of inherited layers, we construct models with different parameter counts and report their Top-1 accuracy in Figure 3(b). MAGNET consistently outperforms all baselines across the entire capacity spectrum. At the largest configuration, it achieves 90.02% Top-1 accuracy, exceeding NKD and MGD by 0.97% and 3.46%, respectively. As model size decreases, all methods experience accuracy degradation, but MAGNET shows the slowest decline. At the smallest configuration, it retains 76.84%, while competing methods fall below or near 76.5%. This demonstrates the method's ability to maintain competitive performance under tight parameter constraints. These results validate the effectiveness of MAGNET's gradient-informed inheritance and adaptive pruning. Unlike static or heuristic designs, it dynamically selects salient structures aligned with parameter availability. This aligns with the intra-layer saliency patterns observed in Figure 2, confirming that task-relevant components can be preserved even under aggressive compression.

## 5 CONCLUSION

We propose MAGNET, a novel knowledge distillation framework that optimizes student model architecture by leveraging gradient-guided saliency profiling. By systematically identifying and selecting the most informative layers, MAGNET achieves superior performance with fewer parameters, outperforming traditional heuristic methods and NAS-based approaches. Experimental results demonstrate that MAGNET offers a powerful, computationally efficient solution for knowledge distillation, maintaining high accuracy while minimizing model complexity.

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

# A APPENDIX

We provide the pseudocode for the complete MAGNET pipeline as follows.

---

**Algorithm 1** MAGNET Pipeline

---

1: **Input:** Teacher model $M_T$, dataset $\mathcal{D}$, student depth $N_S$, global masking ratio $\gamma_{\text{prop}}$, bounds $\gamma_{\min}, \gamma_{\max}$, small constant $\varepsilon$, weights $\alpha, \beta, \gamma_{\text{loss}}$

2: **Coarse-Grained Inheritance**
3: **for** each teacher layer $L_i$ **do**
4:     Compute saliency: $S(L_i) = \frac{1}{|\Theta_i|} \sum_{\theta_k \in \Theta_i} \left| \nabla_{\theta_k} \mathcal{L}_{\text{CE}} \right|$
5: **end for**
6: $\mathcal{L}_T^{\text{inherit}} \leftarrow \text{TopK}(S(L_i), N_S)$
7: Construct student $M_S$ by copying layers in $\mathcal{L}_T^{\text{inherit}}$ to form $\Theta_S$

8: **Fine-Grained Masking**
9: Compute layer saliency $S^{(\ell)}$ for each student layer $L_\ell$ (e.g., by reusing the mean absolute gradients from the inheritance stage)
10: $S_{\min} \leftarrow \min_\ell S^{(\ell)}, \quad S_{\max} \leftarrow \max_\ell S^{(\ell)}$
11: **for** each layer $\ell \in M_S$ **do**
12:     $\tilde{S}^{(\ell)} \leftarrow \dfrac{S^{(\ell)} - S_{\min}}{S_{\max} - S_{\min} + \varepsilon}$
13:     $\gamma_{\text{raw}}^{(\ell)} \leftarrow \gamma_{\max} - (\gamma_{\max} - \gamma_{\min}) \cdot \tilde{S}^{(\ell)}$
14: **end for**
15: $\bar{\gamma}_{\text{raw}} \leftarrow \dfrac{1}{N_S} \sum_\ell \gamma_{\text{raw}}^{(\ell)}$
16: **for** each layer $\ell \in M_S$ **do**
17:     $\gamma_\ell \leftarrow \text{clip}\left( \gamma_{\text{raw}}^{(\ell)} \cdot \gamma_{\text{prop}} / (\bar{\gamma}_{\text{raw}} + \varepsilon), 0, 1 \right)$
18:     $S_p^{(\ell)} \leftarrow \left\{ \left| \nabla_{\theta_k} \mathcal{L}_{\text{CE}} \right| : \theta_k \in \Theta_\ell \right\}$
19:     Sort $S_p^{(\ell)}$ ascending: $S_{(k)}^{(\ell)}$
20:     $q_\ell = \lfloor \gamma_\ell \cdot |\Theta_\ell| \rfloor$
21:     $\tau_{\text{mask}}^{(\ell)} = S_{(q_\ell)}^{(\ell)}$
22:     $\Theta_{\text{mask}}^{(\ell)} = \{ \theta_k \in \Theta_\ell \mid S_p(\theta_k) < \tau_{\text{mask}}^{(\ell)} \}$
23: **end for**
24: $\Theta_{\text{mask}} = \bigcup_\ell \Theta_{\text{mask}}^{(\ell)}$

25: **Training**
26: **for** each minibatch $(x, y) \in \mathcal{D}$ **do**
27:     $z_T \leftarrow M_T(x), \quad z_S \leftarrow M_S(x)$
28:     Extract intermediate features $f_j^T, f_j^S$
29:     Compute:

$$\mathcal{L}_{\text{CE}} = \text{CrossEntropy}(z_S, y)$$
$$\mathcal{L}_{\text{KL}} = \text{KL}(z_S, z_T)$$
$$\mathcal{L}_{\text{Feat}} = \frac{1}{N_S} \sum_{j=1}^{N_S} \begin{cases} 1 - \cos(f_j^S, f_j^T), & j < N_S \\ 1 - \cos(f_{N_S}^S, f_{\text{last}}^T), & j = N_S \end{cases}$$

30:     $\mathcal{L}_{\text{total}} = \alpha \mathcal{L}_{\text{CE}} + \beta \mathcal{L}_{\text{KL}} + \gamma_{\text{loss}} \mathcal{L}_{\text{Feat}}$
31:     Backpropagate: compute $\nabla_{\theta_S} \mathcal{L}_{\text{total}}$
32:     Apply masking: $\nabla_{\theta_j} \leftarrow 0$ if $\theta_j \in \Theta_{\text{mask}}$
33:     Update parameters $\theta_S$
34: **end for**
35: **Return:** Trained student model $M_S$

---

# B  APPENDIX

To validate the practical efficiency of MAGNET, we conduct a benchmark study using the `pynvml` library to capture real-time power usage on a single NVIDIA H100 GPU, under the same training configuration as in our main experiments. Unless otherwise specified, all distillation runs are performed for 50 epochs.

## B.1  PROFILING OVERHEAD VS. DISTILLATION TRAINING

We first analyze the computational cost of the gradient profiling step relative to the full distillation process. For clarity, profiling metrics are reported in seconds (s) and Joules (J), whereas training metrics are reported in hours (h) and kilowatt-hours (kWh). Table 6 summarizes the breakdown for MAGNET on both CIFAR-100 and Food-101. The results indicate that the profiling step incurs negligible overhead, taking approximately **5.77 s** and consuming only **780.06 J** of energy. Notably, even on the larger Food-101 dataset, the profiling cost remains effectively constant because it is performed on a fixed subset of 256 samples. Relative to the 50-epoch training duration, this step accounts for less than **0.04%** of the total wall-clock time and **0.03%** of the total energy, confirming that MAGNET introduces virtually no computational burden compared to the efficiency gains of the derived student architecture.

Table 6: MAGNET's efficiency on CIFAR-100 and Food-101 (scaled to 50 epochs).

| Dataset | Profiling Phase | | Distillation Phase (50 Epochs) | | Ratio (Profile / Total) | |
|---|---|---|---|---|---|---|
| | Time (s) | Energy (J) | Time (h) | Energy (kWh) | Time (%) | Energy (%) |
| CIFAR-100 | 5.77 | 780.06 | 4.74 | 0.74 | **0.03%** | **0.03%** |
| Food-101 | 5.62 | 940.28 | 7.11 | 1.01 | **0.02%** | **0.03%** |

## B.2  COMPARISON WITH DISTILLATION BASELINES

We further compare the total computational cost of MAGNET (profiling plus 50-epoch distillation) against strong fixed-architecture baselines, FitNet Romero et al. (2015) and Attention Transfer (AT) Zagoruyko & Komodakis (2016), on the CIFAR-100 dataset. The results are summarized in Table 7. MAGNET is noticeably more efficient than AT, reducing total training time by approximately **12.01%** (saving over 0.64 hours) and lowering energy consumption by **4.59%**. Moreover, despite incorporating an additional profiling phase, MAGNET achieves lower total energy consumption than FitNet (**0.74 kWh** vs. 0.76 kWh). The overall wall-clock time remains comparable (approximately 4.74 hours), with MAGNET retaining a slight advantage due to reduced FLOPs induced by fine-grained parameter masking.

Table 7: Total efficiency comparison on CIFAR-100 (50 epochs). MAGNET values include both the profiling step and the distillation training. Time and energy savings are computed relative to AT. Lower is better. All values are rounded to two decimal places.

| Method | Total Time (h) | Total Energy (kWh) | Time Savings | Energy Savings |
|---|---|---|---|---|
| AT Zagoruyko & Komodakis (2016) | 5.38 | 0.77 | - | - |
| FitNet Romero et al. (2015) | 4.75 | 0.76 | +11.82% | +2.68% |
| **MAGNET (Ours)** | **4.74** | **0.74** | **+12.01%** | **+4.59%** |

