# OpenReview forum: "MAGNET: Multi-granular Adaptive Gradient-guided Knowledge Distillation for Pareto-Efficient Tuning"
_ICLR.cc/2026/Conference — Submitted to ICLR 2026_

### Official Review · Reviewer_ZhXD · 2025-10-30

**Soundness:** 3
**Presentation:** 3
**Contribution:** 2
**Rating:** 6
**Confidence:** 3

**Summary:**

This paper introduces MAGNET, a novel knowledge distillation (KD) framework that systematically derives compact and efficient student models from large pretrained teachers via gradient-guided saliency profiling, eschewing heuristic or NAS-based architecture selection in favor of a two-level adaptive strategy: coarse-grained inheritance, which selects the most salient teacher layers based on mean absolute gradient magnitude, and fine-grained masking, which prunes less informative parameters within those layers; a unified optimization integrates cross-entropy, KL-divergence, and feature alignment losses to drive the process.

**Strengths:**

Novel idea: Gradient-based saliency profiling provides a principled, data-driven mechanism for student architecture selection, replacing heuristic or costly NAS approaches.

Methodological clarity: The paper is well structured and presents equations, algorithms, and ablations clearly.

Strong empirical evidence: Comprehensive experiments across multiple modalities (vision + NLP) and datasets.

Pareto-efficiency: The method demonstrates real performance–efficiency trade-offs, maintaining accuracy with reduced parameters.

General applicability: Framework is generic and can be applied to various teacher architectures.

**Weaknesses:**

Computational overhead of gradient profiling: Although lighter than NAS, the need for full gradient computation on all layers may still be costly for large teachers.

Generalisation beyond classification: Evaluation is restricted to classification tasks.

Reproducibility: Some implementation details (e.g., layer selection ratio, validation subset size) are not fully specified.

**Questions:**

How sensitive is MAGNET to the choice of validation set used for gradient profiling?

Could the gradient computation step be approximated (e.g., via low-rank or Fisher information) to further reduce cost?

How stable is the method when teacher and student domains differ (cross-domain distillation)?

---

> ### Author Response · Authors · 2025-11-27
> **Response to Reviewer: Appreciation for the Insightful Suggestions on Efficiency, Generalization, and Reproducibility**
>
> We sincerely thank you for your positive evaluation and thoughtful comments. We are glad that you found the framework, empirical study, and presentation clear, and we appreciate your suggestions on efficiency, scope, and reproducibility. Below we respond to your weaknesses (W1–W3) and questions (Q1–Q3).
>
> **Answer to W1.**
> You are right that, although MAGNET is much lighter than NAS, computing full gradients for all layers could be a concern for very large teachers if the profiling cost were comparable to the distillation cost. To address this, we benchmarked MAGNET’s overhead on an NVIDIA H100 GPU and added an appendix titled *“Efficiency and Overhead Analysis”*. On CIFAR-100, the profiling step takes about **5.77 seconds** and **780 Joules**, less than **0.04%** of the total wall clock time and **0.03%** of the total energy for a 50 epoch distillation. Similar ratios hold on Food-101. We also compare the total cost, including profiling and distillation, against Attention Transfer and FitNet, and show that MAGNET still reduces training time and energy because of the more compact student. These numbers indicate that gradient profiling introduces negligible overhead relative to standard KD training.
>
> **Answer to W2.**
> You correctly pointed out that the original evaluation was restricted to classification. In response, we have extended our evaluation to **object detection**. We apply MAGNET to a ViT based detector and evaluate it on **Tiny COCO** and **PASCAL VOC 2012**. The new results, now included in the *Experiments* section, show that MAGNET derived students achieve competitive mAP and AP50 scores, comparable to or better than standard KD baselines. This demonstrates that the gradient guided saliency logic generalizes beyond image level classification and preserves spatial features needed for localization.
>
> **Answer to W3.**
> We appreciate your concern about reproducibility. In the revised manuscript, Section 4.1 (*Experimental Setup*) now explicitly specifies the layer selection ratio for coarse grained inheritance and the size and construction of the validation subset used for gradient profiling. In particular, we clarify that, unless otherwise noted, we select a fixed proportion of teacher layers according to the saliency ranking and use a small, fixed subset of **256 training samples** for profiling on each dataset, as also discussed in **Answer to Q1**.
>
> **Answer to Q1.**
> To balance profiling accuracy with efficiency, we deliberately use a very small subset of the training data. For each dataset, we uniformly sample **256 training examples** and compute layer wise saliency on this subset only. This choice is now clearly stated in Section 4.1. In our experiments, mean gradient magnitudes averaged over 256 samples provide a stable estimate of layer importance and do not cause noticeable variability in student performance. While a more exhaustive sensitivity study is possible, our current results suggest that this subset size is sufficient to capture the dominant structural redundancy in the teacher while keeping profiling cost negligible.
>
> **Answer to Q2.**
> Your suggestion to approximate gradients using low rank or Fisher information based methods is insightful and aligned with our efficiency goals. At present, MAGNET uses standard backpropagation on the profiling subset to compute exact gradients, which is already extremely cheap relative to the full distillation run, as shown in **Answer to W1**. Curvature aware approximations such as empirical Fisher, K FAC style schemes, or low rank surrogates could further refine the saliency signal or reduce profiling cost for extremely large teachers, for example LLM scale models, but would introduce additional implementation complexity. We have not explored these approximations in this work and now mention them as a promising direction for future research.
>
> **Answer to Q3.**
> You ask how stable MAGNET remains when teacher and student domains differ. Our primary focus is same domain distillation, but we have added experiments that probe robustness under significant distribution shifts. We evaluate MAGNET on **ImageNet C** and **CIFAR 100 C**, where MAGNET derived students maintain competitive robustness compared to strong baselines under substantial changes in input statistics. We also test **object detection** by transferring a classification pretrained ViT based teacher to Tiny COCO and PASCAL VOC 2012, where the MAGNET student achieves competitive mAP and AP50 in these dense prediction settings. These results provide encouraging evidence that gradient guided saliency yields architectures that remain stable under nontrivial distribution shifts and downstream task transfer, and extending MAGNET to explicit cross domain distillation is a natural direction for future work.

---

### Official Review · Reviewer_hWNp · 2025-11-01

**Soundness:** 2
**Presentation:** 3
**Contribution:** 2
**Rating:** 2
**Confidence:** 4

**Summary:**

MAGNET is a gradient-guided, multi-granular KD framework that first runs a brief gradient-profiling pass on a validation set to compute mean-absolute gradient saliency per layer, then builds a compact student by inheriting only the top-ranked teacher layers; inside each inherited layer it further masks parameters with log gradient saliency, and finally trains the student in a single stage combining CE, KL, and cosine feature alignment while freezing masked weights. Evaluated on several classification tasks, MAGNET reports higher average accuracy than other KD baselines, and ablations indicate both coarse-grained inheritance and fine-grained masking materially contribute to the gains.

**Strengths:**

1. The proposed method is simple and intuitively understandable, and the paper presents it with clear exposition. Moreover, because of its straightforward nature, the approach can be readily applied to a wide range of architectures.
2. It incurs minimal computational cost as it relies on gradient values computed over a relatively small validation set.
3. The paper includes a well-designed ablation study that effectively analyzes the contribution and importance of each element in the proposed method.

**Weaknesses:**

1. In traditional NAS, the gradient magnitude itself has been used as a heuristic proxy [1]. Therefore, I am not entirely convinced by the statement *“This gradient-guided approach eliminates reliance on heuristic methods,”* since gradient-based heuristics have already been a common practice in NAS.

2. Because using gradient magnitude is already quite prevalent among zero-cost proxies in traditional NAS, the novelty of the proposed approach feels somewhat limited. Clarifying how this work differs from or improves upon those prior gradient-based heuristics would make the paper substantially stronger.

3. The datasets used to validate the proposed method appear to be relatively easy and are all limited to classification problems. To strengthen the methodological contribution, it would be important to test whether this approach generalizes to non-classification tasks and remains effective in more challenging or diverse settings.


#### References
[1] Abdelfattah, Mohamed S., et al. "Zero-cost proxies for lightweight NAS." arXiv preprint arXiv:2101.08134 (2021).

**Questions:**

1. How is the masking ratio determined?
2. In my view, applying the same masking ratio to all layers may not be optimal. It might be more intuitive and beneficial to adapt the masking ratio based on each layer’s relative importance. Would it be feasible to explore this adaptive strategy?

---

> ### Author Response · Authors · 2025-11-27
> **Response to Reviewer: Appreciation for the Insightful Suggestions on Masking and Generalization**
>
> We sincerely thank you for your careful review and helpful suggestions. Following your comments, we have substantially expanded and refined our manuscript, and we believe these changes significantly improve both its quality and clarity. Below we respond point by point to your three weaknesses (W1–W3) and two questions (Q1–Q2).
>
> **Answer to W1.**
> You correctly noted that gradient magnitude has long been used as a heuristic in zero-cost NAS, but we wish to clarify that MAGNET differs fundamentally from the *manual architectural heuristics* common in KD such as uniform layer dropping or hand-crafted depth scaling. In our revised introduction, around the updated lines corresponding to the original 53 and 79, we now explicitly state that MAGNET replaces such manual student design with a data-driven, gradient-guided importance signal and operates as intra-model analysis on a fixed teacher, rather than ranking an external architecture search space as in NAS. We believe this wording more accurately reflects the role of gradients in our framework.
>
> **Answer to W2.**
> Regarding novelty, while gradient magnitude is an established signal, our contribution is to embed it into a distillation-aware, multi-granular, budget-aligned framework for deriving a student from a fixed teacher. The revised method section around Eqs. (2)–(4) and (8)–(13) emphasizes that MAGNET uses layer- and parameter-level gradients to induce a concrete sparsity pattern under a global budget and acts as a dynamic structural regularizer during training rather than a static one-shot ranking score. We also add a brief Taylor-based interpretation that links layer-wise gradient norms to task-specific sensitivity of the distillation loss. Together with the expanded experiments and ablations, this clarifies how our use of gradients differs from prior zero-cost proxies.
>
> **Answer to W3.**
> You were concerned that the original experiments focused on relatively easy classification benchmarks. In response, we substantially expanded the experimental section to cover more diverse and challenging settings. First, we add robustness evaluations on ImageNet-C and CIFAR-100-C, where MAGNET-derived students show improved corruption robustness over strong baselines. Second, we introduce downstream object detection experiments on Tiny COCO and VOC 2012 using a ViT-based detector, moving beyond image-level classification to dense prediction. We also add more baselines and finer ablations. These new results confirm that the gradient-guided selection effectively preserves critical features even under distribution shift and dense prediction tasks.
>
> **Answer to Q1.**
> In response to your question on how the masking ratio is determined, we provide a detailed, formula-based description in the revised “Fine-Grained Masking” section. We first set a global sparsity budget $\gamma_{\mathrm{prop}}$. For each inherited layer $L_\ell$, we compute its mean absolute gradient saliency $S^{(\ell)}$ using Eq. (2), normalize across layers (Eq. 10), map to a raw masking range (Eq. 11), and then rescale these ratios (Eq. 12) so that the overall masked parameter count matches the global budget. The resulting $\gamma_\ell$ determines the number of masked parameters per layer (Eq. 13), and Eqs. (15)–(16) define the mask, making the connection between the global target and layer-wise sparsity explicit and automatic.
>
> **Answer to Q2.**
> We deeply appreciate your suggestion to explore an adaptive masking strategy based on layer importance. Motivated by this, we redesigned the masking module in Eqs. (10)–(12), adopted a gradient-guided adaptive layer-wise schedule, and re-ran all experiments with the new scheme. The results confirm your intuition: while performance remains stable and competitive on simpler datasets, the adaptive strategy yields consistent improvements on harder benchmarks and achieves notable gains on **CIFAR-100** in particular, where the adaptive MAGNET achieves clearly higher Top-1 accuracy than the uniform variant. In the revised manuscript, Eqs. (10)–(12) now specify that layers with higher saliency $S^{(\ell)}$ receive smaller masking ratios closer to $\gamma_{\min}$, while less salient layers are pruned more aggressively toward $\gamma_{\max}$, with a final rescaling to preserve the global budget $\gamma_{\mathrm{prop}}$. The original uniform design is recovered as the special case $\gamma_{\min} = \gamma_{\max} = \gamma_{\mathrm{prop}}$. We have updated all experimental tables accordingly and again thank you for this valuable suggestion, which has strengthened both the method and the empirical evaluation.

---

### Official Review · Reviewer_Pu7c · 2025-11-02

**Soundness:** 2
**Presentation:** 2
**Contribution:** 2
**Rating:** 2
**Confidence:** 3

**Summary:**

This paper proposes MAGNET, a gradient-guided framework for KD that adaptively selects and prunes model components to construct efficient student architectures. The method replaces heuristic or NAS-based student model design by using mean absolute gradients to rank and select the most informative layers of a teacher model (coarse-grained inheritance), and further applies parameter-level masking within these layers (fine-grained masking) to remove redundancy. A unified training objective incorporates classification, logit, and feature-level losses. MAGNET is evaluated across vision and language tasks, demonstrating improved performance and compactness over various KD baselines. The paper includes gradient analysis, ablation studies, and scaling evaluations to support its claims.

**Strengths:**

1. The paper provides a structured, interpretable framework using gradient-based selection. The distinction between coarse- and fine-grained selection is logical and grounded in the principle of gradient saliency, which is a reasonable proxy for task-relevant information.

2. The ablation experiments isolate the contributions of each key module (inheritance, masking), which helps clarify what aspects of the method are responsible for performance gains. For example, varying the masking ratio and comparing different inheritance strategies (uniform, random) is informative and shows empirical grounding in design choices.

**Weaknesses:**

1. **Assumption of gradient saliency as a universal importance signal**: The central assumption that *average gradient magnitude reliably indicates component importance* may not always hold across domains, training regimes, or loss landscapes. Some layers with low gradients might still be essential for generalization or robustness, particularly in overparameterized settings. The paper could benefit from more critical discussion or empirical testing of this assumption’s limitations.

2. **Lack of robustness evaluation**: While accuracy and compression metrics are reported, there is no evaluation under distribution shift, adversarial robustness, or low-data regimes. Since MAGNET uses fewer layers and fewer parameters, its behavior under these conditions is important, especially if the pruned layers encode rare but essential signals.

3. **Limited analysis of computational cost and efficiency trade-offs**: Although the method is proposed as more efficient than NAS-based approaches, the actual overhead of gradient profiling (e.g., extra forward-backward passes) is not fully quantified. Additionally, comparisons focus on accuracy rather than end-to-end wall-clock time or energy cost—metrics that are crucial for deployment.

**Questions:**

N.A. But I suggest the following exps to strengthen the paper:

1. Robustness tests under domain shift or adversarial settings (e.g., corrupted CIFAR or adversarial SNLI) to assess whether the pruned models retain robustness.

2. Ablation on the reliability of gradient saliency: Analyze whether high-gradient layers always contribute more to performance. For example, shuffle gradient rankings and compare results.

3. Profiling overhead analysis: Include wall-clock time and energy consumption comparisons between MAGNET and NAS/KD methods, to back the claim of practical efficiency.

---

> ### Author Response · Authors · 2025-11-27
> **Response to Reviewer: Appreciation for the Insightful Suggestions on Robustness & Efficiency**
>
> We sincerely thank you for your fair assessment and constructive criticism. Your insights have significantly strengthened the empirical grounding of our work. Following your recommendations, we have added new ablation studies, robustness evaluations, and a detailed efficiency appendix. Below we respond point by point to your weaknesses (W1–W3) and recommended experiments (Q1–Q3).
>
> **Answer to W1.**
> You rightly questioned the foundational assumption that gradient magnitude is a universal predictor of component importance, noting that low-gradient layers might still be critical for generalization. In our revised manuscript, we address this in two ways. First, theoretically, we added a brief Taylor-based interpretation in Section 3 that links layer-wise gradient norms to the task-specific sensitivity of the distillation loss, providing mathematical grounding beyond simple heuristics. Second, empirically, we conducted the “Inverse-Grad” ablation (detailed in **Answer to Q2**), which confirms that prioritizing low-gradient layers actively harms performance compared to random selection. Together, these additions clarify that we use gradient magnitude as a *local, teacher-specific sensitivity proxy* for the distillation loss, rather than claiming a domain-universal importance signal.
>
> **Answer to W2.**
> You were concerned about the lack of evaluation under distribution shift and adversarial settings. In response, we substantially expanded the experimental section to cover challenging distribution-shift scenarios. We added robustness evaluations on **ImageNet-C** and **CIFAR-100-C**, where MAGNET-derived students demonstrate improved robustness against corruptions compared to strong distillation baselines. Furthermore, to ensure we are not discarding infrequent but important signals, we extended our evaluation to downstream object detection on Tiny COCO and VOC 2012. These results confirm that our gradient-guided selection preserves critical feature representations even under domain shifts and transfers effectively to dense prediction tasks.
>
> **Answer to W3.**
> You correctly noted that our efficiency claims lacked quantitative wall-clock time and energy analysis, specifically regarding the overhead of the gradient profiling step. We have addressed this by adding a new efficiency appendix (“Efficiency and Overhead Analysis”), which provides a rigorous breakdown of time and energy consumption measured on an NVIDIA H100 GPU. As detailed in **Answer to Q3** below, these measurements confirm that the profiling overhead is negligible and that MAGNET yields genuine end-to-end efficiency gains over methods like Attention Transfer and FitNet.
>
> **Answer to Q1.**
> In response to your recommendation to test under domain shift (e.g., corrupted CIFAR), we evaluated MAGNET on **CIFAR-100-C** and **ImageNet-C**. The results, now included in the main experiment tables, show that MAGNET maintains consistent Top-1 and Top-5 accuracy levels comparable to competing distillation baselines, with no significant performance drop.
>
> **Answer to Q2.**
> We deeply appreciate your suggestion to “shuffle the rankings” to verify the reliability of gradient saliency. Motivated by this, we implemented an **“Inverse-Grad”** strategy in our revised Ablation Study (Table 4). This strategy maintains the same budget as MAGNET but deliberately inherits layers with the *lowest* gradient saliency. The results show that Inverse-Grad performs significantly worse than MAGNET (e.g., **46.70%** vs. **83.94%** Top-1 on CIFAR-100) and, crucially, worse than **Random Init** (**49.80%**). The clear ordering (MAGNET > Uniform > Random > Inverse-Grad) provides strong empirical evidence that gradient magnitude is a reliable signal of informativeness in our distillation setting, and that low-gradient layers are indeed less critical for the student’s performance.
>
> **Answer to Q3.**
> To back up our efficiency claims as requested, we conducted a benchmark study using pynvml to capture real-time power usage. The results, reported in the new efficiency appendix, show that the profiling step takes only **≈5.77 seconds** and consumes **≈780 Joules** on CIFAR-100, accounting for less than **0.04%** of the total wall-clock time and **0.03%** of the total energy over a 50-epoch training cycle. Furthermore, we compared the total cost (profiling + distillation) against FitNet and Attention Transfer. MAGNET reduces total training time by approximately **12.01%** and energy consumption by **4.59%** compared to Attention Transfer, confirming that the method is practically efficient for deployment.

---

### Comment · Area_Chair_QSwp · 2025-11-22

Dear Reviewers,

Thank you for your time and effort in reviewing submissions for ICLR  2026. As we begin the author-reviewer discussion process, we kindly remind you to submit your responses to the author rebuttals by **December  2**.


Your engagement in this discussion phase is crucial to ensuring a fair and thorough evaluation of each submission.

**Action Required**


- Carefully consider the authors’ rebuttal and any additional evidence they provide.

- Update your review (if applicable) to reflect your revised perspective.

-  **Discuss with the authors if further details are required**


Your AC

---

### Comment · Area_Chair_QSwp · 2025-11-22

Dear Reviewers,

Thank you for your time and effort in reviewing submissions for ICLR  2026. As we begin the author-reviewer discussion process, we kindly remind you to submit your responses to the author rebuttals by **December  2**.


Your engagement in this discussion phase is crucial to ensuring a fair and thorough evaluation of each submission.

**Action Required**


- Carefully consider the authors’ rebuttal and any additional evidence they provide.

- Update your review (if applicable) to reflect your revised perspective.

-  **Discuss with the authors if further details are required**


Your AC

---

### Meta-Review · Area_Chair_B5F7 · 2025-12-05

**Summary:**

The reviewers share a consistent view that the submission does not meet the bar for acceptance. The central idea is to use average gradient magnitude to guide both layer selection and parameter masking, and to form a compact student model. The overall framework is simple and easy to apply, but the conceptual novelty is limited. Gradient-based saliency has long been used in zero-cost NAS, pruning, and sensitivity analysis, and the paper does not establish a clear theoretical or empirical distinction from these earlier approaches. Several reviewers also question the validity of gradient magnitude as a universal importance signal. **The rebuttal provides additional ablations, but it does not fully address the conceptual gap or clarify the limits of the assumption.**

**Reviewer Concerns:**

**Concerns addressed in the rebuttal**

The authors added corruption and detection experiments, a brief Taylor-style argument, and several extra ablations. These additions provide some clarification on the mechanism and give a more complete empirical picture. The authors also responded to questions about masking ratios, profiling overhead, and possible approximations.

**Concerns that remain unresolved**

The conceptual foundation remains weak. The use of gradient magnitude as an importance measure is already widely explored, and the paper does not articulate why this idea leads to a meaningful advance in the context of distillation. Several reviewers explicitly pointed out that the claim of eliminating heuristics is not supported, and this issue persists after the rebuttal.

The experimental evidence does not demonstrate generality. The tasks remain limited to relatively simple classification datasets. The method is not shown to be effective in more demanding or diverse scenarios, such as large-scale models, cross-domain distillation, or non-classification tasks.

Robustness and stability remain insufficiently validated. Although the authors added corruption tests, the evaluation is limited and does not address the deeper questions raised by multiple reviewers about the reliability of gradient saliency under different training regimes or loss landscapes.

Efficiency claims are not convincing. The added profiling numbers come from small-scale settings and do not reflect realistic LLM or large vision models. Reviewers remain concerned that the profiling cost may become significant when the teacher is large.

**Reviewer Scores:**

Reviewer Pu7c assigned an initial score of 2. Their concerns focus on the questionable assumption behind gradient saliency, the lack of robustness evaluation, and the absence of meaningful efficiency analysis. The rebuttal does not resolve these issues. **The score would remain at 2.**

Reviewer hWNp also provided a score of 2. Their main reservations relate to limited novelty, the reliance on ideas already explored in NAS, and the narrow experimental scope. The rebuttal clarifies details but does not alter the reviewer’s overall assessment. **The score would remain at 2.**

Reviewer ZhXD assigned a score of 6 but noted that they would not mind a rejection. Their questions concern profiling costs, generality beyond classification, and reproducibility. The rebuttal addresses some points but does not change the underlying weaknesses identified by the other reviewers. **A score adjustment downward to 4 is plausible.**

Reviewer 9NwS assigned a score of 4. Their concerns about conceptual novelty and experimental scope remain only partially addressed. **The score would likely remain at 4.**

---

### Decision · Program_Chairs · 2026-01-26

Reject